# Preparation of Water-in-Oil Nanoemulsions Loaded with Phenolic-Rich Olive Cake Extract Using Response Surface Methodology Approach

**DOI:** 10.3390/foods11030279

**Published:** 2022-01-20

**Authors:** Seyed Mehdi Niknam, Mansoore Kashaninejad, Isabel Escudero, María Teresa Sanz, Sagrario Beltrán, José M. Benito

**Affiliations:** Department of Biotechnology and Food Science (Chemical Engineering Section), University of Burgos, Plaza Misael Bañuelos s/n, 09001 Burgos, Spain; snx1002@alu.ubu.es (S.M.N.); mkx1002@alu.ubu.es (M.K.); iescuder@ubu.es (I.E.); tersanz@ubu.es (M.T.S.); beltran@ubu.es (S.B.)

**Keywords:** water-in-oil (W/O) nanoemulsion, response surface methodology, high-energy emulsification, phenolics, encapsulation, stability

## Abstract

In this study, we aimed to prepare stable water-in-oil (W/O) nanoemulsions loaded with a phenolic-rich aqueous phase from olive cake extract by applying the response surface methodology and using two methods: rotor-stator mixing and ultrasonic homogenization. The optimal nanoemulsion formulation was 7.4% (*w/w*) of olive cake extract as the dispersed phase, and 11.2% (*w/w*) of a surfactant mixture of polyglycerol polyricinoleate (97%) and Tween 80 (3%) in Miglyol oil as the continuous phase. Optimum results were obtained by ultrasonication for 15 min at 20% amplitude, yielding W/O nanoemulsion droplets of 104.9 ± 6.7 nm in diameter and with a polydispersity index (PDI) of 0.156 ± 0.085. Furthermore, an optimal nanoemulsion with a droplet size of 105.8 ± 10.3 nm and a PDI of 0.255 ± 0.045 was prepared using a rotor-stator mixer for 10.1 min at 20,000 rpm. High levels of retention of antioxidant activity (90.2%) and phenolics (83.1–87.2%) were reached after 30 days of storage at room temperature. Both W/O nanoemulsions showed good physical stability during this storage period.

## 1. Introduction

Water-in-oil (W/O) emulsions and nanoemulsions are widely used systems in foods, medicines, and cosmetics for the encapsulation and delivery of bioactive compounds [1,2,3]. W/O nanoemulsions consist of nanosized water droplets dispersed in an oil phase through the action of emulsifiers, which may appear transparent or translucent because of their narrow droplet size distribution [4]. W/O emulsions and nanoemulsions can be used for the encapsulation of amino acids [5], iron [6], gallic acid [7], and other bioactive compounds [8], but also as fat replacers in the development of reduced-fat solid and liquid food formulations [8,9,10,11]. This last application has generated increased interest in recent years to meet consumer demands.

Polyphenols are natural organic compounds with strong antioxidant properties that are mainly found in fresh and processed herbs, fruits, and vegetables [12,13], making them the most promising antioxidants for use in food products. The solid olive mill residue generated during the olive oil production process is an important source of polyphenols with a high level of antioxidant activity, such as hydroxytyrosol [14,15,16]. The solid residue is usually treated to recover the remaining oil. The final defatted solid residue can then be valorized by recovering the polyphenols. These compounds are prone to degradation due to adverse environmental conditions, but their stability and bioavailability can be improved by encapsulation [17,18,19], mainly in single emulsions [20,21,22], double emulsions [23,24,25], or in emulsion gels [11].

As a general trend, the smaller the droplet size, the more stable the W/O nanoemulsions and the longer their shelf life. Nevertheless, formulation and operating conditions for their preparation must be optimized to improve their encapsulation, delivery properties, and their long-term storage stability [8]. Response surface methodology (RSM) is an effective tool for process optimization, thereby minimizing the number of experiments required in order to achieve optimization. It has been applied in several studies examining the effects of different variables in emulsion formulations and their stability [23,26,27,28,29]. In this study, we prepared W/O nanoemulsions loaded with a phenolic-rich aqueous phase from olive cake extract by applying the response surface methodology. The nanoemulsion formulation was first optimized, and then operating conditions to obtain W/O nanoemulsions by two high-energy emulsification methods (rotor-stator mixing and ultrasonic homogenization) were examined. Stability and phenolic release properties of the optimally formulated W/O nanoemulsions over time were also studied.

## 2. Materials and Methods

### 2.1. Materials

Olive cake, the solid residue from a two-phase cold oil extraction process using olives from olive trees of the variety “Serrana de Espadán”, was kindly provided by Cooperativa de Montán (Castellón, Spain). 

Miglyol 812 (Sasol GmbH, Hamburg, Germany), polyglycerol polyricinoleate (PGPR, Brenntag AG, Essen, Germany), and Tween 80 (polyoxyethylene sorbitan monooleate, Sigma-Aldrich, Darmstadt, Germany), were used as an oil, lipophilic, and hydrophilic emulsifier, respectively.

Sodium carbonate, ethanol, gallic acid, DPPH, and Trolox were purchased from Sigma-Aldrich (Darmstadt, Germany). Folin-Ciocalteu reagent (VWR International Eurolab, Barcelona, Spain) and hydrochloric acid (37%, Acros Organics, Geel, Belgium) were also used for nanoemulsion characterization. Milli-Q water (Millipore, St. Louis, MO, USA) was used throughout this study.

### 2.2. Phenolic-Rich Olive Cake Extract Used as the Aqueous Phase

The phenolic extract was obtained from the ground and defatted olive cake by ultrasound-assisted extraction using a 50% *v/v* ethanol–water solution as a solvent, as was explained in a previous study [16]. This extract had a total phenolic content (TPC) of 17.27 ± 0.87 mg gallic acid equivalents per gram of dried and defatted olive cake and 228.2 ± 0.4 mg Trolox/L of antioxidant activity (AA). The main phenolic compounds in this extract were hydroxytyrosol and tyrosol, followed by oleuropein and its derivatives [16]. Other compounds such as flavonoids were also present at low concentrations [15,16,30,31]. It was stored at −20 °C before use as the aqueous phase of the nanoemulsions.

### 2.3. Preparation of Water-in-Oil (W/O) Nanoemulsions

Emulsions were formulated by mixing Miglyol 812 oil with PGPR and Tween 80 emulsifiers. Due to limited amounts of the olive cake extract, the experiments generated by the RSM were performed using 50% *v/v* ethanol-water solution as the aqueous phase. The dispersed phase was added dropwise to the continuous phase and the mixture was stirred for 5 min at 500 rpm. The emulsification was carried out using a high-intensity ultrasonic homogenizer (Sonics VCX500, 500 W, 20 kHz, Newtown, CT, USA) for 10 min effective time, in 5 s pulses (5 s off and 5 s on), with a 40% amplitude [23]. Then, emulsification of the optimized formulation was performed using the phenolic-rich olive cake extract as the aqueous phase and two high-energy emulsification methods: (i) high-speed homogenization using a Miccra D9 with a DS-5/K-1 rotor-stator (ART Prozess & Labortechnik, Mülheim, Germany); (ii) high-intensity ultrasonic homogenization using the aforementioned homogenizer.

### 2.4. Characterization of W/O Nanoemulsions

The nanoemulsions were characterized by the following methods described in previous studies [16,21,23,32,33] and explained in the Appendix A: (i) droplet size distribution, mean droplet diameter, and polydispersity index (PDI) by dynamic light scattering using a Zetasizer Nano ZS (Malvern Instruments Ltd., Malvern, U.K.); (ii) TPC using the Folin-Ciocalteu standard method; (iii) AA by the DPPH method [34]; (iv) TPC and AA retained within the aqueous phase, following the method developed by Regan and Mulvihill [23,35]. All measurements were performed at least three times.

The droplet size distribution of all nanoemulsions was measured 1 and 30 days after preparation, stored at 4 °C and at room temperature, in darkness, to evaluate their stability. Optical characterization of each optimally formulated nanoemulsion was performed for 30 days at 25 °C by static multiple light scattering using a Turbiscan Lab Expert (Formulaction Co., L’Union, France) [23,36,37].

### 2.5. Experimental Design

The response surface methodology (RSM) was used to evaluate the effects of variable factors (X_i_), each with 3 levels, on the response variable (Y, droplet size). A central composite design (CCD) with two replicates of the central point was used in this study. 

The following second-order polynomial model, Equation (1), was used to predict the variation in the response variable (Y):(1)$$Y=c_{0}+\underset{i}{\sum }c_{i}X_{i}+\underset{\mathrm{ii}}{\sum }c_{\mathrm{ii}}X_{i}^{2}+\underset{\mathrm{ij}}{\sum }c_{\mathrm{ij}}X_{i}X_{j}$$
where X_i_ and X_j_ are the independent variables; c_0_ is a constant; and c_i_, c_ii_, and c_ij_ are the coefficients of the linear, quadratic, and interactive terms, respectively.

Aqueous phase content, surfactant content, and the HLB (hydrophilic-lipophilic balance) number were the factors studied for nanoemulsions formulation with minimum droplet size, with 16 experimental runs. The HLB number was fixed for each experiment, based on a pre-calculation of the necessary content of each emulsifier in the surfactant mixture to be prepared [38]. Subsequently, two experimental designs with two factors and 10 experimental runs each were applied to obtain the optimal operating conditions for the two high-energy emulsification methods. The factors and the levels of the three experimental designs are summarized in Appendix A.

Analysis of variance (ANOVA) and the least significant difference (LSD) test were applied to determine the effect of the factors on the response. The significance of the models was evaluated through values of the statistical parameters *F* and *p* (probability), with a confidence level of 95% (*p* < 0.05), using Statgraphics Centurion 18 software (Statgraphics Technologies, Inc., Warrenton, VA, USA).

## 3. Results and Discussion

### 3.1. W/O Nanoemulsion Formulation

Table 1 shows the effect on droplet size (and on PDI) of the three factors used in the CCD for the optimization of the W/O nanoemulsion formulation. The coefficient of determination (R^2^) for particle size was 0.907.

ANOVA (Table 2) showed that three coefficients (c_1_, c_3_, and c_13_) of the quadratic model, Equation (1), were statistically significant (*p* < 0.05). The HLB number (X_3_) and the aqueous phase content (X_1_) had a stronger effect on droplet size than the surfactant content. The significance of the HLB number and the surfactant content and their effect on particle size were previously reported [38,39].

Response surface and contour plots were generated to study the effect of the independent variables on the droplet size of the W/O emulsions. Figure 1a shows that, regardless of the surfactant content, a decrease in the HLB number from 11 to 3 resulted in a droplet size decrease, because PGPR is more efficient than Tween 80 in the formation of nanoemulsions with a smaller droplet size.

Figure 1b shows the effect of the HLB number and the aqueous phase concentration on droplet size at a fixed surfactant content (11% *w/w*). Particle size decreased significantly as the aqueous content increased for high HLB numbers (7–11). However, aqueous content did not have a significant effect on droplet size for low HLB numbers (3–7). Moreover, it can be observed in Figure 1c that, for a fixed HLB number of 7, the surfactant content had no considerable effect on particle size.

Numerical optimization was performed using a desirability function method. The optimal operating conditions for the emulsification of the phenolic extract are those leading to a stable nanoemulsion with minimum droplet size, according to Stokes’ law [40]. An optimum formulation with 98.6% desirability was predicted to be achieved by combining 11.2% *w/w* of surfactant content with an HLB number of 3.36 and 7.4% *w/w* of aqueous phase content.

Three replicates with the optimum conditions (81.42% oil, 11.19% surfactant, and 7.4% aqueous phase) were performed to confirm the result, and a nanoemulsion was obtained with a droplet diameter of 132.9 ± 16.7 nm and a narrow droplet size distribution (PDI = 0.153 ± 0.134).

### 3.2. Effects of High-Energy Emulsification Methods on W/O Nanoemulsion Droplet Size

As explained in previous studies, droplet size mainly depends on the type of oil and emulsifier used, under suitable homogenization conditions [41,42,43]. Two high-energy methods (rotor-stator mixing and ultrasonic homogenization) were studied to optimize the operating conditions needed to obtain stable W/O nanoemulsions with the aforementioned optimum formulation and with a minimum droplet size.

Table 3 shows the effect on particle size and PDI of the factors used in the CCD for the optimization of the W/O nanoemulsion formulation using each method. The coefficients of determination (R^2^) for particle size were 0.909 and 0.872 for rotor-stator mixing and ultrasonic homogenization, respectively. These R^2^ values, which were higher than 0.85, indicate that the models adequately represented the relationship within the chosen parameters.

ANOVA (Table 4) showed that all regression coefficients of the model, Equation (1), were statistically significant (*p* < 0.05), with a greater effect of time (X_2_) than the rotation speed (X_1_) on particle size. Moreover, no coefficient was statistically significant for the ultrasonic homogenizer model, in which amplitude (X_2_) had significantly stronger effect on the droplet size than time (X_1_).

Figure 2a shows the response surface plot with the effects of time and rotation speed on droplet size of nanoemulsions prepared with the rotor-stator mixer: an increase in processing time at higher rotation speeds (20,000–29,000 rpm) caused an increase in droplet size, while at lower speeds (11,000–20,000 rpm), the opposite effect took place and particle size decreased significantly.

The interaction of time and amplitude on emulsion droplet size in nanoemulsions prepared with the ultrasonic homogenizer is shown in Figure 2b: time variation did not have a noticeable effect on droplet size at lower amplitudes. However, as mentioned above, the amplitude is the most significant factor in ultrasound-assisted emulsification experiments. As reported in other studies [27,44,45], for shorter time periods (5–10 min), increasing the amplitude has a negative effect on emulsion droplet size, while for longer time periods (10–15 min), no considerable effect was observed by varying the amplitude. 

An optimum formulation with 100% desirability was predicted to be reached by rotor-stator mixing at 20,000 rpm and 10.1 min to obtain a W/O nanoemulsion with a particle size of 103.1 nm and a PDI of 0.211, while experimental results were 105.8 ± 10.3 nm and 0.255 ± 0.045 for average particle size and PDI, respectively (Table 3). These results confirm the reliability of the rotor-stator mixing experimental design. 

The situation was slightly different using ultrasonic homogenization: a droplet size of 63.31 nm with a PDI of 0.422 was predicted to be obtained by emulsification for 5 min at 20% amplitude with 100% desirability. However, the results observed using these operating conditions were 71.09 ± 7.35 nm and 0.480 ± 0.283 for average particle size and PDI, respectively (Table 3). The lack of visual stability, droplet size growth in the initial days, and the wide distribution of particle size (PDI) for the nanoemulsion obtained using these operating conditions caused us to conclude that emulsification at 15 min and 20% amplitude are the optimal operating conditions when using an ultrasonic homogenizer. An O/W nanoemulsion with 104.9 ± 6.7 nm and 0.156 ± 0.085 for average particle size and PDI, respectively, was achieved with these factor levels, resulting in a considerably lower PDI with a satisfactory droplet size diameter.

### 3.3. Characterization of Optimal W/O Nanoemulsion

The long-term stability of the optimized W/O nanoemulsions was evaluated by two methods: (i) droplet size distribution measurement at 25 °C and 4 °C after 1 day and 30 days from emulsification; (ii) monitoring of the evolution of backscattering (BS) profiles over time at room temperature to detect emulsion destabilization.

Figure 3a,b shows no significant changes in the average droplet size of optimally formulated nanoemulsions, with no apparent variations in visual stability after 30 days of storage. No change was observed in the droplet size distribution of the optimal nanoemulsion prepared with the rotor-stator mixer (Figure 3a), despite a slight PDI increase during the storage period at room temperature. However, a lower temperature had a better effect on its stability and caused a considerably lower PDI with the same particle size. 

The droplet size distribution for the optimal nanoemulsion prepared with the ultrasonic homogenizer (Figure 3b) indicates no significant differences between storage conditions at both temperatures.

Stability measurements of optimal nanoemulsions obtained by monitoring the backscattering (BS) profiles for 30 days at 25 °C are shown in Figure 4. BS profiles for the optimal nanoemulsion prepared with the rotor-stator mixer (Figure 4a) showed sedimentation at the bottom of the sample and creaming after the first week of storage, with a slight emulsion destabilization at the end of the storage period. Regarding the optimal nanoemulsion prepared with the ultrasonic homogenizer, the corresponding BS profiles (Figure 4b) showed the same levels of sedimentation and creaming after 30 days of storage as those of the nanoemulsion prepared with the rotor-stator mixer after the first week of analysis. Additionally, it showed a slight increase in particle size in the last days of the storage period. However, the optimal nanoemulsions were physically stable; visually, there was no indication of any destabilization; and no phase separation was observed throughout the storage period. W/O nanoemulsions formulated in this study were more stable than double emulsions prepared using the same high-energy emulsification methods in a previous study [23].

Free polyphenols are easily degraded and W/O nanoemulsions may be effective in overcoming this problem. In this study, freshly extracted phenolic compounds from olive cake had a TPC of 17.27 ± 0.87 mg gallic acid/g olive cake and 228.2 ± 0.4 mg Trolox/L of AA. After a 30 day storage period, 87.2% and 83.1% retentions of TPC were recorded for the optimal nanoemulsions prepared with the rotor-stator mixer and the ultrasonic homogenizer, respectively, whereas a 90.2% AA retention was recorded for both methods, indicating good chemical stability of the W/O nanoemulsions. 

These experimental results are similar to those obtained for phenolic encapsulation in W/O emulsions in food applications. For instance, in the study performed by Rabelo et al. [46], 2% berry extract rich in anthocyanins was encapsulated in food-grade W/O nanoemulsions: 94.6% of total anthocyanin content retention was observed after 30 days of storage. Gomes et al. [7] studied gallic acid encapsulation in single emulsions formulated with soybean oil, with more than 85% gallic acid retention after 7 days of storage, especially when using W/O emulsions.

## 4. Conclusions and Future Research Needs

The formulation and preparation of W/O nanoemulsions loaded with phenolic-rich olive cake extract were optimized in this study. The optimal formulation was 7.4% (*w/w*) of olive cake extract as the dispersed phase, and 11.2% (*w/w*) of a surfactant mixture of PGPR (97%) and Tween 80 (3%) in Miglyol oil as the continuous phase. The optimal emulsification condition with an ultrasonic homogenizer was obtained by sonication for 15 min at 20% amplitude, which resulted in a W/O nanoemulsion with a 104.9 ± 6.7 nm droplet diameter and a PDI of 0.156 ± 0.085. Furthermore, an optimal nanoemulsion with a droplet size of 105.8 ± 10.3 nm and a PDI of 0.255 ± 0.045 was obtained using a rotor-stator mixer for 10.1 min at 20,000 rpm. After a 30 day storage period at 25 °C, total phenolic content analysis of the optimal nanoemulsions prepared with the rotor-stator mixer and the ultrasonic homogenizer showed 87.2% and 83.1% retention of the initial phenolic compounds, respectively, and a 90.2% retention of antioxidant activity for both nanoemulsions. In spite of slight levels of sedimentation and creaming for the optimal nanoemulsions prepared with each device, they showed satisfactory physical stability and no droplet size growth over a 30 day storage time at room temperature.

These optimal W/O nanoemulsions may be suitable for application in the food industry, but the following research would need to be conducted to ensure suitability: (i) a microbiological analysis of the olive cake extract used as the aqueous phase should be carried out before it is used in food applications; (ii) Miglyol oil can be replaced by most common and healthy edible oils, such as olive oil; (iii) a maximum daily limit of 2.6 mg of PGPR emulsifier per kilogram of body weight is suggested [47]; therefore, natural alternatives, such as lecithins [48,49], are strongly recommended for PGPR substitution in food products [50].

## Figures and Tables

**Figure 1 foods-11-00279-f001:**
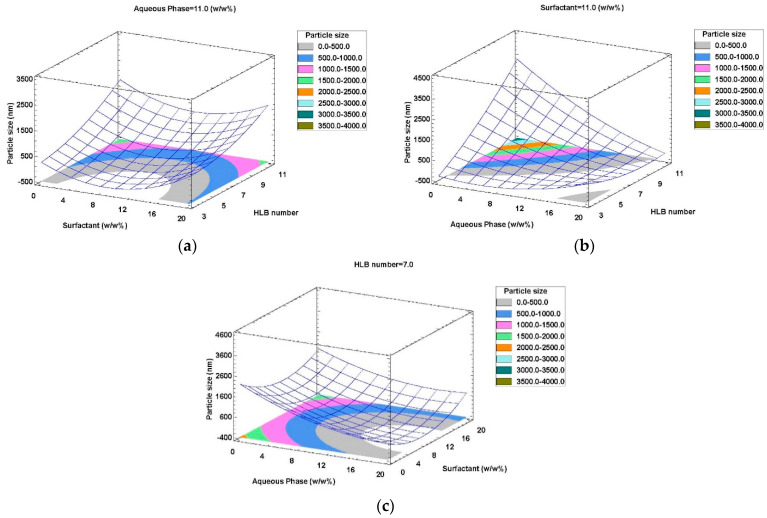
RSM plots of the effects of factors on particle size of W/O emulsions. (**a**) Interaction of surfactant content and HLB number; (**b**) interaction of aqueous phase content and HLB number; (**c**) interaction of surfactant and aqueous phase contents.

**Figure 2 foods-11-00279-f002:**
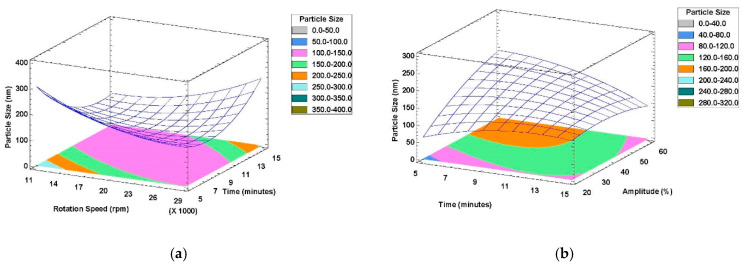
RSM plots of the effects of factors on particle size of W/O nanoemulsions: (**a**) rotor-stator mixer; (**b**) ultrasonic homogenizer.

**Figure 3 foods-11-00279-f003:**
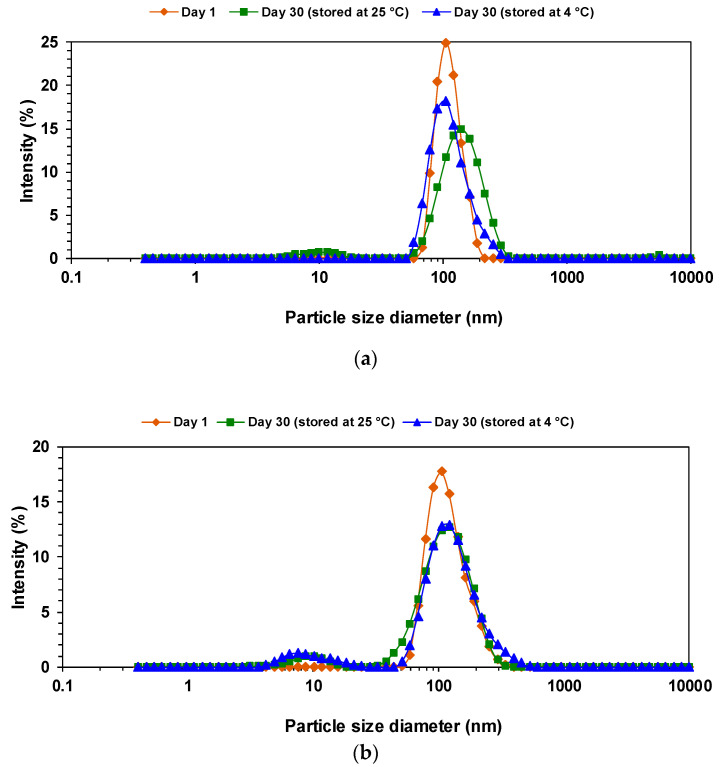
Particle size distribution of optimally formulated W/O nanoemulsions after 1 day and a 30-day storage period at 25 °C and at 4 °C: (**a**) rotor-stator mixer; (**b**) ultrasonic homogenizer.

**Figure 4 foods-11-00279-f004:**
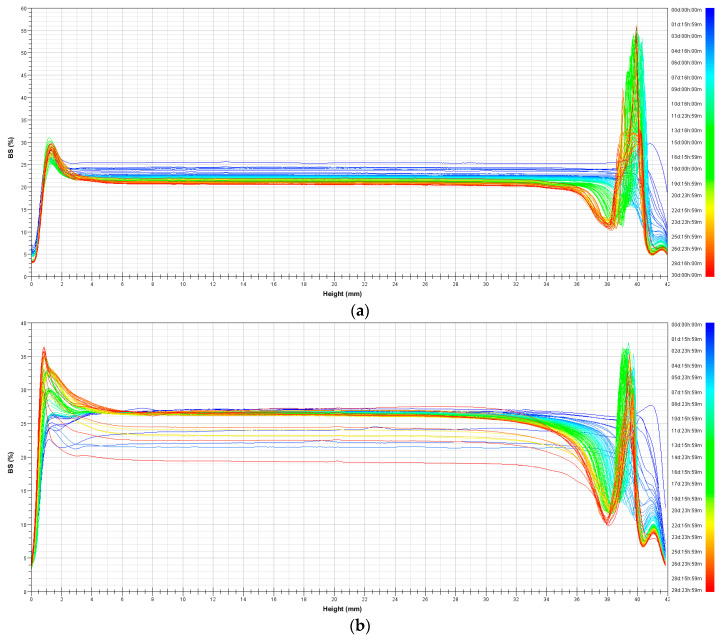
BS profiles of optimally formulated W/O nanoemulsions through 30 days of storage at 25 °C: (**a**) rotor-stator mixer; (**b**) ultrasonic homogenizer.

**Table 1 foods-11-00279-t001:** Matrix of the CCD and experimental data for the W/O emulsion formulation.

Run	Independent Variables			Response Variables			
	Aqueous Phase Content (X_1_, % *w/w*)	Surfactant Content (X_2_, % *w/w*)	HLB Number (X_3_)	Droplet Size (Y, nm)		PDI	
				Mean	SD	Mean	SD
1	2	20	11	3688	464	1	0
2	20	2	3	313.5	12.7	0.956	0.076
3	2	2	3	143.9	24.6	0.094	0.071
4	20	20	11	692.1	121.0	1	0
5	11	20	7	804.2	237.3	1	0
6	2	2	11	3856	589	1	0
7	20	20	3	1152	30	0.164	0.148
8	2	11	7	259.3	51.7	0.246	0.210
9	11	11	3	177.4	47.0	0.104	0.009
10	11	11	7	331.8	116.0	0.082	0.054
11	11	11	7	142.3	13.1	0.222	0.094
12	11	2	7	356.1	68.1	0.350	0.278
13	20	2	11	263.2	71.8	1	0
14	11	11	11	320.3	64.6	0.666	0.413
15	2	20	3	359.8	14.0	0.055	0.061
16	20	11	7	372.1	136.3	0.416	0.303

PDI: polydispersity index. SD: standard deviation.

**Table 2 foods-11-00279-t002:** ANOVA of the regression coefficients (Equation (1)) for the droplet size of the W/O emulsions. Statistically significant terms (*p* < 0.05) are written in bold.

Source	Regression Coefficients	*F*-Ratio	*p*-Value
c_0_	−388.136	-	-
**c_1_**	8.48871	169.34	0.0488
c_2_	−146.709	17.32	0.1501
**c_3_**	244.208	248.00	0.0404
c_11_	4.23006	17.24	0.1505
c_12_	1.88194	10.35	0.1918
**c_13_**	−26.217	396.89	0.0319
c_22_	7.49487	54.11	0.0860
c_23_	−2.75521	4.38	0.2837
c_33_	17.2365	11.17	0.1851

**Table 3 foods-11-00279-t003:** Matrix of the CCD and experimental data for W/O emulsion preparations using two high-energy emulsification methods.

Run	Rotor-Stator Mixer					
	Independent Variables		Response Variables			
	Rotation Speed (X_1_, rpm)	Time (X_2_, min)	Droplet Size (Y, nm)		PDI	
			Mean	SD	Mean	SD
1	20,000	10	105.8	10.3	0.255	0.045
2	20,000	5	223.5	26.8	0.425	0.238
3	29,000	15	280.7	29.0	0.571	0.093
4	29,000	10	133.2	6.6	0.166	0.107
5	20,000	10	105.0	4.7	0.192	0.224
6	11,000	15	138.4	2.2	0.187	0.043
7	29,000	5	136.4	6.1	0.175	0.016
8	11,000	10	148.5	30.0	0.323	0.044
9	11,000	5	286.9	15.2	0.556	0.150
10	20,000	15	112.6	4.4	0.185	0.020
	**Ultrasonic Homogenizer**					
	**Time (X_1_, min)**	**Amplitude (X_2_, %)**	**Droplet Size (Y, nm)**		**PDI**	
			**Mean**	**SD**	**Mean**	**SD**
11	5	20	71.09	7.35	0.480	0.283
12	15	20	104.9	6.7	0.156	0.085
13	10	20	113.2	6.9	0.136	0.109
14	15	40	114.7	18.1	0.103	0.011
15	15	60	111.5	9.9	0.060	0.035
16	10	60	120.3	12.9	0.074	0.018
17	10	40	175.9	3.5	0.163	0.040
18	10	40	188.1	5.2	0.168	0.044
19	5	40	106.5	6.5	0.192	0.107
20	5	60	238.8	12.7	0.019	0.015

PDI: polydispersity index. SD: standard deviation.

**Table 4 foods-11-00279-t004:** ANOVA of the regression coefficients (Equation (1)) for the W/O nanoemulsions prepared by high-energy emulsification methods.

Source	Rotor-Stator Mixer			Ultrasonic Homogenizer		
	Regression Coefficients	*F*-Ratio	*p*-Value	Regression Coefficients	*F*-Ratio	*p*-Value
c_0_	937.701	-	-	−188.939	-	-
c_1_	−0.0362116	287.63	0.0375	31.4482	12.70	0.1742
c_2_	−89.7357	6900.01	0.0077	8.68788	65.80	0.0781
c_11_	0.00000049	11380.94	0.0060	−0.942343	17.40	0.1498
c_12_	0.00162667	66978.00	0.0025	−0.377775	76.71	0.0724
c_22_	2.66829	32446.77	0.0035	−0.0435214	9.50	0.1997

## Data Availability

Data is contained within the article (or Appendix A).

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
