# Peer review of "Preparation of Water-in-Oil Nanoemulsions Loaded with Phenolic-Rich Olive Cake Extract Using Response Surface Methodology Approach"

_foods, 2022, doi:10.3390/foods11030279_

Round 1

Reviewer 1 Report

The writing in the full manuscript needs to be checked and improved, such as tenses, accuracy, and conciseness.

  1. Which is the main theme of this study? The theme and innovation should be ascertained and highlighted. Line 10–12 is confusing for your theme.
  2. Line 16: The full name should be provided for the first occurrence of its abbreviation (PGPR).
  3. In Section 2.2, the composition of phenolic extract should be given.
  4. After storage, the antioxidant activity remained 96%, whereas the phenolic retentions were lower than 90%. Therefore, in addition to polyphenols, is there other composition that showed antioxidant activity? According to the results of Question 3, maybe you can answer this question.
  5. Has the rationality of model fitting been verified? The necessary parameters, such as R2, R2Adj, CV, were not given or discussed.
  6. The similar studies [1] should be discussed in this article, and what are the advantages or progress in this article?

[1] Niknam SM, Escudero I, Benito JM. Formulation and Preparation of Water-In-Oil-In-Water Emulsions Loaded with a Phenolic-Rich Inner Aqueous Phase by Application of High Energy Emulsification Methods. Foods. 2020; 9(10):1411. https://doi.org/10.3390/foods9101411

  1. It is recommended to provide pictures of the nanoemulsions during storage.

Reviewer 2 Report

I reviewed the manuscript (foods-1533448) entitled, Preparation of water-in-oil nanoemulsions loaded with phenolic-rich olive cake extract – A response surface methodology approach. The study optimized the nanoemulsion preparation process for the incorporation of phenolic compounds and the presence of these phenolic compounds further enhanced the physical stability of the formed nanoemulsion. Although the study approach is simple with less novelty, the research findings are interesting and provide ways to enhance the olive cake extract utilization.

  1. Introduction should be improved with the role of optimization and the need of optimization studies. The valorization of Olive Cake for phenolics should be addressed in introduction
  2. Research problem should be addressed
  3. Line 21: What does author mean Satisfactory levels of antioxidant activity?
  4. 1 materials: add chemical supplier city and country
  5. 3. provide the reference for Preparation of water-in-oil (W/O) nanoemulsions
  6. 4. Characterization of W/O nanoemulsions: describe the all methodologies
  7. 5. Experimental design: coded values should be provided in supplementary
  8. Line 116: lesser? It should be Least
  9. Figures (all) resolution should be improved
  10. Overall, manuscript should be revised carefully several times

Reviewer 3 Report

Overall, the introduction, material and methods, results, and discussion, as well as conclusions are properly presented, confirming the high scientific expertise of the research team. This paper investigates the preparation of W/O nanoemulsions loaded with phenolic-rich aqueous phase from olive cake extract using the response surface methodology. The research work topic is important and worth of investigation and approving, however there are many shortcomings which must be rectified. In general, important information is presented.

  1. Title should be corrected: Preparation of Water-in-Oil Nanoemulsions Loaded with Phenolic-Rich Olive Cake Extract using Response Surface Methodology Approach

2. Abstract: the aim is not clearly presented; please start the abstract with “This study investigates”.

  1.  Please do not use abbreviations in the abstract

4.      This works has a variety of data which are not apparent by just reading the abstract. There appears to be some information, which can add to knowledge in this growing field.

5.      Please synthetise keywords!

6.      L33-L47: please give more information and develop!

7.      Table 3: please write with bold the significant (p<0.05) terms.

  1. L41-L42: There are fifteen references for only one sentence! Please revise all these references. “Their stability and bioavailability can be improved by encapsulation in single and double emulsions and emulsion gels [17,20–35]”.
  2. L152-L155: please rephrase this sentence
  3. “L231: Figure 3 should be replaced by Figure 3a and 3b”,
  4. In the discussion, the author would have benefited from a better understanding of the existing literature.
  5. In some sentences, English appears not to be adequate.
  6. To use a space between the number and the unit, as 20 °C; and not to use a space between number and percentage, as 10%, for example.
  7. Conclusion: please include limitations and future research area!
  8. References must be revised.

Round 2

Reviewer 1 Report

Although the authors have carefully addressed all the comments from the reviewers, there are some points still need to be revised.

Line 12  Past tense can be used.

There are 6 paragraphs in the introduction. It should be restructured. Generally, 3 paragraphs are recommended.

L 104 Please specify the which supplementary material (the number of figure or table should be given).

The format of Table 1 should be met the requirement of the publication.

Table 1 the Significant Figures of mean values and SD should be the same.

The total number of references are increased from 60 to 68. For a research paper, no more than 40 or 45 are welcome.

Reviewer 2 Report

The authors have carefully addressed all the comments raised by reviewers and the article quality has been substantially improved for acceptance and eventual publication in FOODS.
